# Novel ANO1 Inhibitor from *Mallotus apelta* Extract Exerts Anticancer Activity through Downregulation of ANO1

**DOI:** 10.3390/ijms21186470

**Published:** 2020-09-04

**Authors:** Yohan Seo, Nguyen Hoang Anh, Yunkyung Heo, So-Hyeon Park, Phan Van Kiem, Yechan Lee, Duong Thi Hai Yen, Sungwoo Jo, Dongkyu Jeon, Bui Huu Tai, Nguyen Hoai Nam, Chau Van Minh, Seung Hyun Kim, Nguyen Xuan Nhiem, Wan Namkung

**Affiliations:** 1College of Pharmacy and Yonsei Institute of Pharmaceutical Sciences, Yonsei University, 85 Songdogwahak-ro, Yeonsu-gu, Incheon 21983, Korea; yohanseo@dgmif.re.kr (Y.S.); ykheo107@naver.com (Y.H.); sso_0605@naver.com (S.-H.P.); llyycc94@naver.com (Y.L.); dsdyu2005@naver.com (S.J.); armisael1990@gmail.com (D.J.); kimsh11@yonsei.ac.kr (S.H.K.); 2Interdisciplinary Program of Integrated OMICS for Biomedical Science Graduate School, Yonsei University, Seoul 03722, Korea; 3New Drug Development Center, Daegu-Gyeongbuk Medical Innovation Foundation, Daegu 41061, Korea; 4Graduate University of Sciences and Technology, VAST, 18 Hoang Quoc Viet, Cau Giay, Hanoi 100000, Vietnam; anhnh@tnus.edu.vn (N.H.A.); phankiem@yahoo.com (P.V.K.); bhtaiich@gmail.com (B.H.T.); 5Graduate Program of Industrial Pharmaceutical Science, Yonsei University, Incheon 21983, Korea; 6Institute of Marine Biochemistry, Vietnam Academy of Science and Technology (VAST), 18 Hoang Quoc Viet, Cau Giay, Hanoi 100000, Vietnam; haiyenk51a@gmail.com (D.T.H.Y.); namnguyenhoai@imbc.vast.vn (N.H.N.); cvminh@vast.vn (C.V.M.)

**Keywords:** *Mallotus apelta*, anoctamin 1, inhibitor, cytotoxicity, apoptosis

## Abstract

Anoctamin1 (ANO1), a calcium-activated chloride channel, is frequently overexpressed in several cancers, including human prostate cancer and oral squamous cell carcinomas. ANO1 plays a critical role in tumor growth and maintenance of these cancers. In this study, we have isolated two new compounds (**1** and **2**) and four known compounds (**3**–**6**) from *Mallotus apelta.* These compounds were evaluated for their inhibitory effects on ANO1 channel activity and their cytotoxic effects on PC-3 prostate cancer cells. Interestingly, compounds **1** and **2** significantly reduced both ANO1 channel activity and cell viability. Electrophysiological study revealed that compound **2** (Ani-D2) is a potent and selective ANO1 inhibitor, with an IC_50_ value of 2.64 μM. Ani-D2 had minimal effect on cystic fibrosis transmembrane conductance regulator (CFTR) chloride channel activity and intracellular calcium signaling. Notably, Ani-D2 significantly reduced ANO1 protein expression levels and cell viability in an ANO1-dependent manner in PC-3 and oral squamous cell carcinoma CAL-27 cells. In addition, Ani-D2 strongly reduced cell migration and induced activation of caspase-3 and cleavage of PARP in PC-3 and CAL-27 cells. This study revealed that a novel ANO1 inhibitor, Ani-D2, has therapeutic potential for the treatment of several cancers that overexpress ANO1, such as prostate cancer and oral squamous cell carcinoma.

## 1. Introduction

Anoctamin1 (ANO1), also called transmembrane protein 16A (TMEM16A), is known as a calcium-activated chloride channel (CaCC) [1,2,3]. ANO1 is widely expressed in various tissues and regulates many physiological activities such as epithelial cell secretion, smooth muscle contraction, cell growth, and nerve cell transmission [3,4,5]. In particular, ANO1 is highly amplified and expressed in various carcinomas including oral squamous cell carcinoma (OSCC), prostate, breast, and esophageal cancers and is involved in the proliferation, metastasis, and invasion of cancer cells [6,7,8,9,10].

Although the underlying mechanism is unclear, anticancer effects of ANO1 inhibition have been reported in several studies [11,12,13]. The cell proliferation, metastasis, and invasion of prostate and oral cancer cells were significantly reduced by the inhibition of ANO1 channel function and reduction of ANO1 protein level [6,10]. In addition, downregulation of ANO1 with the treatment of ANO1 shRNA significantly decreased tumor growth in a prostate cancer xenograft mouse model [6]. Emerging evidence suggests that ANO1 inhibitors could be a feasible approach to treat prostate cancer, oral cancer, and various other carcinomas that express high levels of ANO1. To date, several ANO1 inhibitors have been reported, including CaCC_inh_-A01 [14], T16A_inh_-A01 [15], idebenone [12], tannic acid [16], Ani9 [17], and Ani9-5f [13]. However, their mechanism of action and pharmacological properties have not been fully understood, and these inhibitors are in the early stages of drug discovery.

Natural products have been the source of various medicinal preparations which have shown pharmacological potential as therapeutic agents in a variety of carcinomas [18]. *Mallotus apelta* (Lour.) Müll. Arg (Euphorbiaceae) has been used in oriental medicine. The compounds from this genus showed many interesting biological activities including antiviral [19], hepatoprotective [20], and cytotoxic activities [21,22,23]. The cytotoxic properties are attributed to the presence of specific chromenes. Herein, we reported the isolation and structural elucidation of two new chromenes and four known compounds from *M. apelta.* These compounds were evaluated for cytotoxic activity and ANO1 inhibitory effects. We further investigated the physiological effects of compound **2** and the cytotoxic mechanism in prostate cancer and oral squamous cancer cell lines.

## 2. Results

### 2.1. Isolation of Compounds

The leaves of *M. apelta* were sonicated with methanol to yield methanol extract. The methanol extract was suspended in water and then partitioned with dichloromethane to yield dichloromethane (MA1) residue and water layer (MA2). Using various chromatographic methods, two new chromenes and four known compounds were isolated and elucidated (Figure 1).

### 2.2. Structural Elucidations of Compounds

Compound **1** was isolated as a yellow powder. Its molecular structure was deduced as C_20_H_18_O_5_ by a quasi-molecular ion peak at *m*/*z* 339.1227 [M + H]^+^ (calcd. for C_20_H_19_O_5_, 339.1232) in the high-resolution electrospray ionisation mass spectrometry (HR-ESI-MS) and in conjunction with NMR data. The ^1^H-NMR spectrum of **1** showed signals corresponding to an AA′BB′ azomatic ring (*δ*_H_ 7.52 and 6.86 (each, 2H, d, *J* = 8.5 Hz)), two *trans* olefinic protons (*δ*_H_ 8.00 and 7.70 (each, 1H, d, *J* = 15.5 Hz)), two *cis* olefinic protons (*δ*_H_ 6.61 and 5.52 (each, 1H, d, *J* = 10 Hz)), and two methyl groups (*δ*_H_ 1.56 (6H, s)). The ^13^C-NMR spectrum of **1** revealed signals of 20 carbon atoms which were classified by HSQC into nine non-protonated carbons, nine methines, and two methyl groups (Table 1). A de-shielded carbon signal (*δ*_C_ 194.1) was assigned for a ketone group. The heteronuclear multiple bond correlation (HMBC) spectrum of **1** exhibited H-C interactions of two individual structure fragments (Figure 2). The *J* coupling constant value between H-2′ and H-3′ (*J*_H-2′/H-3′_ = 15.5 Hz) together with HMBC correlations between H-3′ (*δ*_H_ 7.70) and C-1′ (*δ*_C_ 194.1)/C-4′ (*δ*_C_ 128.4)/C-5′ (*δ*_C_ 131.2)/C-9′ (*δ*_C_ 131.2), H-5′ (*δ*_H_ 7.52)/H-9′ (*δ*_H_ 7.52) and C-7′ (*δ*_C_ 161.2) indicated the presence of the *trans*-*p*-coumaroyl group. Meanwhile, the *J* coupling constant value between H-3 and H-4 (*J*_H-3/H-4_ = 10.0 Hz) together with HMBC correlations between H-4 (*δ*_H_ 6.61) and C-3 (*δ*_C_ 125.3)/C-10 (*δ*_C_ 104.0)/C-5 (*δ*_C_ 157.7)/C-9 (*δ*_C_ 161.6), H-8 (*δ*_H_ 5.93) and C-6 (*δ*_C_ 106.7)/C-7 (*δ*_C_ 167.5)/C-9 (*δ*_C_ 161.6)/C-10 (*δ*_C_ 104.0) suggested the presence of a 2*H*-chromene moiety. The HMBC correlations between H-11 (*δ*_H_ 1.56)/H-12 (*δ*_H_ 1.56) and C-2 (*δ*_C_ 79.1)/C-3 (*δ*_C_ 125.3) demonstrated two methyl groups at C-2. The other two de-shielded carbons (C-5 (*δ*_C_ 157.7) and C-7 (*δ*_C_ 167.5)) suggested the presence of hydroxyl groups at C-5 and C-7. Finally, direct C-C bonding between C-6 and C-1′ was shown to form a ketone group at C-1′ which consisted of the molecular formula of **1**. Consequently, the chemical structure of **1** was established and named as malloapelta I.

Compound **2** was isolated as a yellow amorphous powder. HR-ESI-MS analysis of **2** indicated its molecular formula to be C_20_H_20_O_6_, showing a quasi-molecular ion peak at *m*/*z* 357.1332 [M + H]^+^ (calcd. for C_20_H_21_O_6_, 357.1338). The ^1^H-NMR spectrum of **2** contained signals of an AA′BB′ aromatic proton ring [*δ*_H_ 7.31 and 6.87 (each, 2H, d, *J* = 8.5 Hz)], a *cis* olefinic proton [*δ*_H_ 6.62 and 5.50 (each, 1H, d, *J* = 10.5 Hz)], an oxymethine group [*δ*_H_ 5.34 (1H, dd, *J* = 3.0 and 13.0 Hz)], a methylene group [*δ*_H_ 3.07 (1H, dd, *J* = 13.0 and 17.0 Hz) and 2.78 (1H, dd, *J* = 3.0 and 17.0 Hz)], and two methyl groups [*δ*_H_ 1.44 and 1.43 (each, 3H, s)]. The ^13^C-NMR spectrum of **2** revealed signals of 20 carbon atoms and was divided by HSQC into nine non-protonated carbons (*δ*_C_ 196.1, 162.5, 162.3, 161.2, 158.4, 130.5, 103.1, 102.8, and 78.4), eight methine carbons [*δ*_C_ 128.0 × 2, 126.3, 115.7 × 2, 115.3, 96.3, and 78.9)], one methylene carbon (*δ*_C_ 43.1), and two methyl carbons (*δ*_C_ 28.5 and 28.4) (Table 1). Basically, the ^1^H- and ^13^C-NMR data of **2** showed strong similarity to those of **1**. Differences were found at signals of H-2′/C-2′ and H-3′/C-3′. Particularly, signals for a *trans*-double bond in compound **1** were replaced by signals of an oxymethine group (*δ*_H_ 5.34 and *δ*_C_ 78.9) and a methylene group (*δ*_H_ 3.07 and 2.78 and *δ*_C_ 43.1) in **2**. The HMBC correlations between H-3′ (*δ*_H_ 5.34) and C-1′ (*δ*_C_ 196.1)/C-4′ (*δ*_C_ 130.5)/C-5′/C-9′ (*δ*_C_ 128.0) indicated the presence of a hydroxyl group at C-3′. Thus, methylene was assigned at C-2′, which was also confirmed by the HMBC correlation between H_2_-2′ (*δ*_H_ 3.07 and 2.78) and C-4′ (*δ*_C_ 130.5). Since the optical rotation of **2** was approximately zero, it was suggested to occur in racemic solution. This deduction was agreed by the lack of Cotton effect observed in the electronic circular dichroism (ECD) spectrum of **2**. Consequently, the chemical structure of **2** was established and named malloapelta II. Isolation of pure enantiomer of **2** was attempted on preparative HPLC using a chiral column. Unfortunately, our isolation process failed because a small amount of compound **2** was obtained.

The known compounds were elucidated as malloapelta B (**3**) [24], apigenin-7-*O*-β-d-glucopyranoside (**4**) [25], blumenol C glucopyranoside (**5**) [26], and acantrifoside E (**6**) [27] by comparing their observed and reported physical data (Figure 1).

### 2.3. Identification and Characterization of Novel ANO1 Inhibitors

A cell-based assay was performed to identify novel ANO1 inhibitor from the methanol extract of *Mallotus apelta*. The inhibitory effect of compounds **1**–**6** on ANO1 activity was measured with yellow fluorescent protein (YFP-F46L/H148Q/I152L) quenching assay in Fisher rat thyroid (FRT) cells expressing human ANO1. As shown in Figure 3A, both compounds **1** (Ani-D1) and **2** (Ani-D2) inhibited ANO1 activity >99% at the concentration of 25 μM. Interestingly, Ani-D2 decreased ANO1 protein levels more strongly than Ani-D1 in PC-3 cells (Figure 3B,C). To investigate whether Ani-D2 affects the mRNA expression of ANO1, we performed quantitative real-time PCRs in PC-3 cells. As shown in Figure 3D, Ani-D2 did not affect the ANO1 mRNA expression up to 10 µM.

### 2.4. Cytotoxic Effect of Compounds on PC-3 Cells

All compounds were evaluated for cytotoxic activity on PC-3 prostate cancer cells at a concentration of 30 μM. Compounds **1**–**3** potently inhibited cell viability, with IC_50_ values of 8.89 ± 0.17, 7.29 ± 0.20, and 1.60 ± 0.05 μM, respectively, compared to the positive control, capecitabine (Table 2 and Appendix A). Ani-D2 was chosen for further study because Ani-D2 strongly blocked ANO1 channel activity and reduced ANO1 protein expression levels and cell viability in PC-3 cells.

### 2.5. Selective Inhibition of ANO1 by Ani-D2

To investigate the inhibitory effect of Ani-D2 on ANO1 chloride channel activity, apical membrane currents were measured in FRT cells expressing human ANO1. Pretreatment with Ani-D2 significantly inhibited ATP-activated ANO1 chloride current, with an IC_50_ value of 2.64 μM (Figure 4A,B). To observe the effect of Ani-D2 on calcium signaling, FRT cells were loaded with a fluorescent calcium indicator, Fluo-4. Pretreatment of Ani-D2 did not significantly alter the ATP-induced increases in cytosolic calcium concentration (Figure 4C). To investigate the effect of Ani-D2 on other chloride channel activity, we measured the apical membrane currents of cystic fibrosis transmembrane conductance regulator (CFTR) in FRT cells expressing human wild-type CFTR. Ani-D2 exhibited a minimal inhibitory effect on CFTR channel activity at a concentration of 30 μM, showing complete inhibition of ANO1 (Figure 4D).

### 2.6. Inhibitory Effects of Ani-D2 on Cell Proliferation and Migration in PC-3 and CAL-27 Cells

Previous studies have shown that pharmacological blockade of ANO1 inhibits cell proliferation of metastatic prostate cancer cells and metastasis of oral cancer cells [10,28]. As shown in Figure 5A,B, Ani-D2 induced a significant reduction in ANO1 protein levels in a dose-dependent manner. In addition, Ani-D2 significantly reduced cell viability in PC-3 and CAL-27 cells, and the cytotoxic effects were minimal in ANO1 KO cells (Figure 5C,D). Notably, Ani-D2 did not affect cell viability of ANO1 KO PC-3 cells at a concentration of 3 μM, showing significant cytotoxic effects on ANO1 expressing PC-3 cells. Ani-D2 also strongly decreased cell viability in ANO1 expressing CAL-27 but not in ANO1 KO CAL-27 cells. To investigate the cytotoxic effect of Ani-D2 on non-cancerous cells, we observed the cytotoxicity of Ani-D2 in HaCaT cells, an immortal keratinocyte cell line. Ani-D2 showed low toxicity in HaCaT cells expressing ANO1 compared with PC-3 and CAL-27 cells up to 3 μM (Appendix A).

To verify whether Ani-D2 inhibits migration of the cancer cells, a cell wound healing assay was performed on PC-3 and CAL-27 cells highly expressing ANO1. Ani-D2 at concentrations of 1, 3, and 10 μM strongly inhibited the migration in PC-3 cells by 31.3, 68.0, and 88.3%, respectively, and in CAL-27 cells by 31.8, 64.0, and 87.5%, respectively (Figure 6).

### 2.7. Increase in Caspase-3 Activity and PARP Cleavage by Ani-D2 in PC-3 and CAL-27 Cells

Pharmacological inhibition of the ANO1 protein causes apoptosis in various cancer cells [28]. To investigate whether Ani-D2 induces apoptosis in PC-3 and CAL-27 cells expressing high levels of ANO1, the effect of Ani-D2 on caspase-3 activity and PARP cleavage was observed in these cells. As shown in Figure 7A,B, Ani-D2 strongly increased caspase-3 positive cells in both PC-3 and CAL-27 cells. Caspase-3 activity was significantly increased by Ani-D2 in a dose-dependent manner in PC-3 and CAL-27 cells, and the Ani-D2 induced increases in caspase-3 activity were fully inhibited by AC-DEVD-CHO, a specific caspase-3 inhibitor (Figure 7C,D). In addition, Ani-D2 treatment significantly increased cleaved PARP-1 in PC-3 and CAL-27 cells (Figure 7E,F).

## 3. Discussion

ANO1 amplification and overexpression have been reported in various carcinomas [7,8,9]. Recent evidence suggests that ANO1 is a potential therapeutic target for cancers such as prostate, oral, breast, and pancreatic cancer [8,10,12]. In this study, we performed cell-based screening to identify a novel ANO1 inhibitor from the methanol extract of *M. apelta* and found that Ani-D2 is a bona fide inhibitor of ANO1. Previous reports suggest that ANO1 inhibitors can modulate cancer progression by downregulation of ANO1 in various cancer cells. ANO1 inhibitors have decreased the cell viability in breast cancer, head and neck squamous cell carcinoma (HNSCC), and esophageal squamous cell carcinoma (ESCC) by inhibiting the activity of Ca^2+^/calmodulin-dependent protein kinase II (CAMKII) and the expression of epidermal growth factor receptor (EGFR) [8,29]. In addition, overexpression of ANO1 promotes tumor growth by activating EGFR-mediated AKT/SRC/ERK1/2 signaling or Ras-Raf-MEK-ERK1/2 signaling pathway [30,31]. Thus, ANO1 may be a potential drug target for cancer therapy, and ANO1 inhibitors may have therapeutic potential for cancer treatment.

*M. apelta* is widely distributed in China and Vietnam and has been used as a traditional medicine to treat chronic hepatitis, leukemia, and colitis. Flavonoids extracted from *M. apelta* leaf have antioxidant activity and inhibit liver fibrosis in a carbon tetrachloride (CCl_4_)-induced fibrosis in rat via modulating TGF-beta/Smad and NF-κB signaling [32]. Benzopyran derivatives from *M. apelta* show cytotoxic effects on human hepatocellular carcinoma (Hep-2) and rhabdosarcoma (RD) cells [33,34]. These results suggest that *M. apelta* extract may contain useful active substances that are potential agents for cancer treatment. Fortunately, in the present study, we found an active substance, Ani-D2, from the methanol extract of *M. apelta* showing potent and selective inhibition of ANO1 (Figure 4). Ani-D2 not only inhibits ANO1 chloride channel activity but also significantly reduces protein levels of ANO1 in PC-3 and CAL-27 cells (Figure 5A,B). Notably, Ani-D2 significantly reduced cell viability of PC-3 and CAL-27 cells expressing ANO1 but showed minimal inhibitory effects on cell viability of ANO1 KO PC-3 and CAL-27 cells at concentrations up to 3 µM (Figure 5C,D). However, Ani-D2 strongly decreased cell viability at 10 µM in ANO1-KO PC-3 and CAL-27 cells. These results suggest that there is a mechanism of action other than downregulation of ANO1.

In the present study, we established ANO1 knockout cell line using ANO1 CRISPR/Cas9, and the cells that grew well without ANO1 were selected during the selection period. Thus, ANO1-KO PC-3 and CAL-27 cells may be different from the original PC-3 and CAL-27 cells expressing high levels of ANO1, and there was no significant difference in doubling time between the original cell lines and the KO cell lines. Thus, the ANO1 KO cell line is not a complete model but is suitable for use in investigating whether a compound affects cell viability or migration through ANO1-dependent manner.

Previously, we reported that idebenone, luteolin, and Ani9-5f significantly reduced both ANO1 channel activity and protein expression levels and showed strong inhibition of cell viability in PC-3 cells expressing high levels of ANO1 [12,13,17,35]. However, kaempferol and Ani9, which potently inhibit only ANO1 channel activity with minimal effect on ANO1 protein expression levels, showed a weak inhibitory effect on cell viability of PC-3 cells [17,35]. These findings indicate that reducing protein expression levels of ANO1 may be more critical in exhibiting anticancer effects than blocking ANO1 channel activity. Thus, Ani-D2 may have a strong inhibitory effect on cell viability of PC-3 cells, because it potently reduces not only the ANO1 channel activity but also the protein expression level of ANO1. In addition, Ani-D2 significantly reduced the cell migration rate of PC-3 and CAL-27 cells expressing ANO1 (Figure 6) and increased caspase-3 activity and PARP cleavage, the emblematic marker of apoptosis (Figure 7). Taken together, our results strongly suggest that Ani-D2 exhibits cytotoxicity in prostate and oral cancer cells by at least partially inducing apoptosis through inhibition of ANO1.

ANO1 knockdown significantly reduced cell proliferation in both tumor cell lines and HaCaT cells [36]. Interestingly, HaCaT cells, a spontaneously immortalized human keratinocyte, and PC-3 prostate cancer cells express similar levels of ANO1 protein; however, ANO1 knockdown more strongly reduced cell proliferation of PC-3 cells compared with HaCaT cells [36]. As shown in Figure 5C and Appendix A, our results are consistent with the previous report. Although further research is needed in the future, these results suggest that downregulation of ANO1 by small-molecule inhibitors may be more harmful to tumor cells than normal cells.

## 4. Materials and Methods

### 4.1. General

All NMR spectra were recorded on a Bruker 500 MHz. Data processing was performed with MestReNova ver. 9.0.1. HR-ESI-MS spectra were obtained using an AGILENT 6550 iFunnel Q-TOF LC/MS system. HPLC was performed using an AGILENT 1200 HPLC system. Column chromatography (CC) was performed on silica-gel (Kieselgel 60, 230–400 mesh, Merck, Darmstadt, Germany) or RP-18 resins (30–50 μm, Fuji Silysia Chemical Ltd., Triangle Park, NC, USA). For thin layer chromatography (TLC), pre-coated silica-gel 60 F_254_ (0.25 mm, Merck) and RP-18 F_254_S (0.25 mm, Merck) plates were used.

### 4.2. Plant Material

The leaves of *Mallotus apelta* (Lour.) Mull.Arg. were collected at Ngoc Thanh, Phuc Yen, Vinh Phuc, Vietnam (21°22′35.4″ N + 105°43′23.9″ E) in August 2018 and identified by Dr. Nguyen The Cuong, Institute of Ecology and Biological Resources, VAST. A voucher specimen (MA1808) was deposited at the Institute of Ecology and Biological Resources, VAST.

### 4.3. Extraction and Isolation

The dried leaves of *M. apelta* (5.0 kg) were sonicated twice with hot methanol (each 10 L, 4 h, 50 °C) and the solvent was removed in vacuo to yield methanol extract (568 g). The methanol extract was suspended in H_2_O (4.0 L) and then partitioned with dichloromethane to yield dichloromethane (MA1, 250.0 g) residue and H_2_O layer (MA2).

The MA1 (120.0 g) fraction was loaded on a silica gel CC and eluted with a solvent system of *n*-hexane‒acetone (40:1, 20:1, 10:1, 5:1, 1:1, *v*/*v*) to give five smaller fractions, MA1A (15.0 g), MA1B (23.0 g), MA1C (15.0 g), MA1D (25.0 g), and MA1E (10.0 g), respectively. MA1B fraction was chromatographed on a silica gel CC, eluting with dichloromethane‒acetone (40:1, *v*/*v*) to give four sub-fractions, MA1B1 (1.0 g), MA1B2 (1.0 g), MA1B3 (1.2 g), and MA1B4 (1.3 g). The MA1B3 fraction was chromatographed on an RP-18 CC, using methanol‒water (3.5:1, *v*/*v*) as the eluent solvent to give three smaller fractions, MA1B1A (100 mg), MA1B1B (400 mg), and MA1B1C (100 mg). Compound **1** was obtained from MA1C1B using an HPLC system (J’sphere column, ODS H-80, 4 µm, 150 length × 20 mm ID) eluting 55% acetonitrile in water with a flow rate of 3 mL/min. The MA1B3 fraction was chromatographed on a silica gel CC, using *n*-hexane‒ethyl acetate (4:1, *v*/*v*) as the eluent solvent to give three smaller fractions, MA1B3A (80 mg), MA1B3B (150 mg), and MA1B3C (100 mg). MA1B3B was chromatographed on an RP-18 CC, using methanol‒water (3.5:1, *v*/*v*) as the eluent solvent to yield **2** (14.8 mg) and **3** (6.7 mg). The H_2_O layer (MA2) was removed from organic solvent and chromatographed on a Diaion HP-20 column eluting with water to remove sugar components, and then the concentration of methanol in water was increased (25, 50, 75, and 100%, *v*/*v*) to obtain four fractions, MA2A, MA2B, MA2C, and MA2D, respectively. The MA2C fraction was loaded on a silica gel CC and eluted with a solvent system of dichloromethane:methanol 20:1, 10:1, 5:1, 1:1 to give four smaller fractions: MA2C1 (500 mg), MA2C2 (700 mg), MA2C3 (1.2 g), and MA2C4 (1.0 g). The MA2C2 fraction was chromatographed on an RP-18 CC, using acetone‒water (1:4, *v*/*v*) as the eluent solvent to give four fractions, MA2C2A (100 mg), MA2C2B (150 mg), MA2C2C (260 mg), and MA2C2D (110 mg). Compound **5** was obtained from MA2C2C using an HPLC system (J’sphere column, ODS H-80, 4 µm, 150 length × 20 mm ID) eluting 22% acetonitrile in water with a flow rate of 3 mL/min. The purification of the MA2C2D using an HPLC system (J’sphere column, ODS H-80, 4 µm, 150 length × 20 mm ID) eluting 25% acetonitrile in water with a flow rate of 3 mL/min yielded **6** (3.0 mg). The MA2C3 fraction was chromatographed on an RP-18 CC, using acetone‒water (1:1.5, *v*/*v*) as the eluent solvent to give five fractions, MA2C3A (120 mg), MA2C3B (150 mg), MA2C3C (300 mg), MA2C3D (50 mg), and MA2C3E (150 mg). The MA2C3C was chromatographed on a Sephadex LH20 CC, using methanol‒water (1:1, *v*/*v*) as the eluent solvent to yield **4** (30.4 mg).

#### 4.3.1. Malloapelta I (**1**, Ani-D1)

Yellow amorphous powder; C_20_H_18_O_5_, HR-ESI-MS *m*/*z*: 339.1227 [M + H]^+^ (calcd. for C_20_H_19_O_5_, 339.1232); ^1^H (CD_3_OD, 500 MHz), and ^13^C NMR (CD_3_OD, 125 MHz) data; see Table 1.

#### 4.3.2. Malloapelta II (**2**, Ani-D2)

Yellow amorphous powder; [α]_D_^25^ 0.0 (c 0.1, CDCl_3_); C_20_H_20_O_6_, HR-ESI-MS *m*/*z*: 357.1332 [M + H]^+^ (calcd. for C_20_H_21_O_6_, 357.1338); ^1^H (CDCl_3_, 500 MHz), and ^13^C NMR (CDCl_3_, 125 MHz) data; see Table 1.

### 4.4. Cell Culture

Fisher rat thyroid (FRT) cells that stably express ANO1 and CFTR were generated as described previously [37]. FRT cells were cultured in Coon’s modified F12 medium with 10% fetal bovine serum (FBS), 2 mM L-glutamine, 100 units/mL penicillin, and 100 μg/mL streptomycin. PC-3 and CAL-27 cells were cultured in RPMI 1640 medium and Dulbecco’s modified Eagle medium (DMEM), respectively, supplemented with 10% FBS, 100 units/ml penicillin, and 100 μg/mL streptomycin.

### 4.5. Construction of ANO1 Knockout (KO) Cells

PLentiCRISPRv2 vector containing Cas9 and CRISPR guide RNA targeting ANO1 (CCTGATGCCGAGTGCAAGTA) (Clone ID: X35909) was purchased from Genscript (Piscataway, NJ, USA). In total, 1500 ng of the CRISPR plasmid, 1200 ng of packaging plasmid (psPAX2), and 400 ng of envelope plasmid (pMD2.G) were co-transfected to HEK293T cells in 6-well plates. The supernatant containing lentiviral particles was collected 48 h post transfection and was filtered by 0.45 μm syringe filter. Cells were treated with the lentiviral particles mixed with fresh medium at 1:1 ratio in 24-well plates overnight. ANO1 knockout cells were then selected by puromycin (Sigma-Aldrich, St. Louis, MO, USA) 72 h after virus transduction.

### 4.6. YFP Fluorescence Quenching Analysis

FRT cells that stably express both YFP variant (YFP-H148Q/I152L/F46L) and ANO1 were plated in 96-well plates at a density of 2 × 10^3^ cells per well. After 48 h incubation, each well was washed twice with PBS. Then, the test compounds mixed in PBS were treated for 20 min. YFP fluorescence of each well was measured every 0.4 s for 5 s by FLUOstar Omega microplate reader. To measure ANO1-mediated iodide influx, 100 μL of 70 mM iodide solution with 100 μM ATP was automatically injected by microplate reader to each well 1 second after the measurement initiation. The inhibitory effect of test compounds on ANO1 activity was measured using the initial iodide influx rate determined from the initial slope of fluorescence decrease after ATP injection.

### 4.7. Short-Circuit Current

Snapwell inserts containing ANO1 and CFTR expressing FRT cells were mounted in Ussing chambers (Physiologic Instruments, San Diego, CA, USA). Basolateral bath was filled with HCO_3_^−^ buffered solution containing (in mM) 120 NaCl, 5 KCl, 1 MgCl_2_,1 CaCl_2_, 10 D-glucose, 2.5 HEPES, and 25 NaHCO_3_ (pH7.4), and apical bath was filled with a half-Cl^−^ solution. In the half-Cl^−^ solution, 65 mM NaCl in the HCO_3_^−^ buffered solution was replaced by Na-gluconate. The basolateral membrane was permeabilized with 250 μg/mL amphotericin B. Cells were bathed for 20 min and aerated with 95% O_2_/5% CO_2_ at 37 °C. ATP was applied to the apical membrane to activate ANO1, and forskolin was applied to the apical membrane to activate CFTR. Then, **2** was applied to both apical and basolateral bath solution 20 min before ANO1 and CFTR activation. Apical membrane currents were measured with an EVC4000Multi-Channel V/I Clamp (World Precision Instruments, Sarasota, FL, USA) and Power Lab 4/35 (AD Instruments, Castle Hill, Australia). Data were analyzed using Lab chart Pro 7 (AD Instruments, Castle Hill, Australia). The sampling rate was 4 Hz.

### 4.8. Intracellular Calcium Measurement

FRT cells were cultured in 96-well black-walled microplates and loaded with Fluo4 NW according to the manufacturer’s protocol (Invitrogen, Carlsbad, CA, USA). Briefly, the cells were incubated with 100 μL assay buffer (1X Hanks’ balanced salt solution with 2.5 mM probenecid and 20 mM HEPES) including Fluo-4 NW. After 1 h incubation, the 96-well plates were transferred to a plate reader for fluorescence assay. Fluo-4 fluorescence was measured with a FLUO star Omega microplate reader (BMG Labtech) equipped with syringe pumps and custom Fluo-4 excitation/emission filters (485/538 nm).

### 4.9. Western Blot Analysis

Sample preparation was performed as described previously [17]. The protein samples were separated by 4–12% Tris-Glycine-PAG Pre-Cast Gel (KOMA BIOTECH) and transferred to polyvinylidene Fluoride (PVDF) membranes. Blocking was done using 5% bovine serum albumin (BSA) in Tris-buffered saline with 0.1% Tween 20 (TBST) for 1 h. Membranes were then incubated with corresponding primary antibodies, including anti-ANO1 (Abcam), anti-β-actin (Santa Cruz Biotechnology), and anti-cleaved PARP (BD Biosciences) antibodies, followed by incubation of HRP-conjugated anti-secondary IgG antibodies (Enzo life science) for 1 h. Finally, visualization was done with the ECL Plus Western Blotting System (GE Healthcare).

### 4.10. Real-Time RT-PCR Analysis

Total mRNA was extracted using TRIzol reagent (Invitrogen, Carlsbad, CA, USA). Total mRNA was reverse-transcribed with random hexamer primers, an oligo (dT) primer, and SuperScript III Reverse Transcriptase (Invitrogen). StepOnePlus Real-Time PCR System (Applied Biosystems, Foster City, Calif) and Thunderbird SYBR qPCR mix (Toyobo, Osaka, Japan) were used for quantitative real-time PCRs. The thermal cycling conditions included an initial step of 95 °C for 5 min followed by 40 cycles of 95 °C for 10 s, 55 °C for 20 s, and 72 °C for 10 s. The primer sequences used were as follows: ANO1 sense, 5′-GGAGAAGCAGCATCTATTTG-3′; ANO1 antisense, 5′-GATCTCATAGACAATCGTGC-3′; size of ANO1 PCR product, 82 base pairs.

### 4.11. Cell Viability Assay

MTS cell proliferation assay was performed using the CellTiter 96^®^ AQueous One Solution Cell Proliferation Assay kit (Promega). PC-3 and CAL-27 cells were cultured with medium supplemented with 3% FBS for 24 h in 96-well plates. Once the cells reached approximately 20% confluence, compounds or vehicle were treated in medium, freshly exchanged every 24 h. Medium was completely removed after 48 h treatment, and MTS assay was done as recommended in the supplier’s protocol. The absorbance of formazan was measured at a wavelength of 490 nm by Infinite M200 microplate reader (Tecan).

### 4.12. Wound Healing Assay

PC-3 and CAL-27 cells were cultured in a 96-well plate to form a monolayer of approximately 100% confluence. Wound area was formed using a 96-Well WoundMaker (Essen BioScience, MI). After wound formation, each well was washed twice with serum free media and incubated with 200 μL of culture medium containing 2% FBS. IncuCyte ZOOM (Essen BioScience, MI) was used to take images of the wounds, and IncuCyte software was used to analyze the percentage of wound closure.

### 4.13. Caspase-3 Activity Assay

PC-3 and CAL-27 cells were cultured in 96-well plates to reach 30% confluence. Then, **2** (Ani-D2) and Ac-DEVD-CHO, a caspase-3 inhibitor, were treated to the corresponding well. After 24 h, each well was washed with PBS and incubated at room temperature with 100 μL of PBS containing 1 μM of NucView 488 caspase-3 substrate. After 30 min incubation, cells were stained with 1 μM Hoechst 33342. The fluorescence of NucView 488 and Hoechst 33342 were each measured by FLUOstar Omega microplate reader (BMG Labtech) and multicolor images were taken via Lionheart FX Automated Microscope (BioTek, Winooski, VT, USA).

### 4.14. Statistical Analysis

All experiments were conducted independently at least three times. The results for multiple trials were presented as the means ± S.E. Statistical analysis was performed using Student’s *t*-test or analysis of variance as appropriate. *p* values less than 0.05 were regarded as statistically significant. GraphPad Prism Software was used to plot the dose–response curve and calculate IC_50_ values.

## Figures and Tables

**Figure 1 ijms-21-06470-f001:**
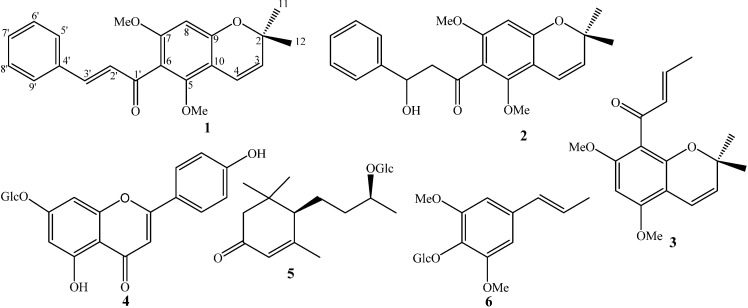
Chemical structures of compounds **1**–**6**.

**Figure 2 ijms-21-06470-f002:**
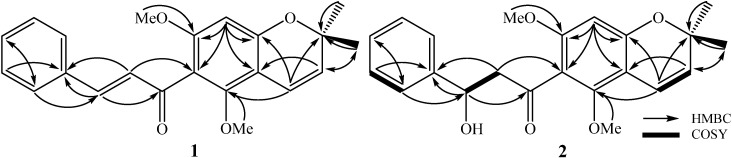
The key HMBC and COSY correlations of compounds **1** and **2.**

**Figure 3 ijms-21-06470-f003:**
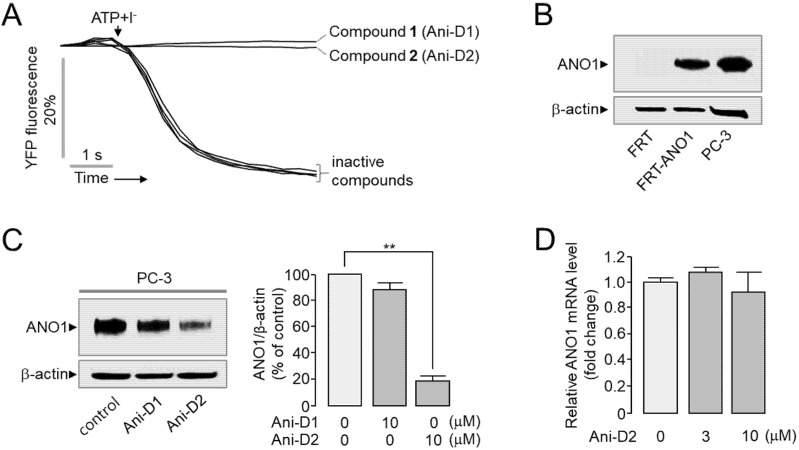
Identification of novel ANO1 inhibitor, Ani-D2. (**A**) Inhibitory effect of compounds **1**–**6** on ANO1 activity was measured in FRT cells expressing human ANO1 and a mutant YFP. The cells were pretreated with compounds **1** (Ani-D1) and **2** (Ani-D2) at 25 µM for 20 min. ANO1 was activated by 100 ATP µM. (**B**) Representative Western blot analysis of ANO1 in FRT, FRT-ANO1, and PC-3 cells (two independent experiments performed). ANO1 protein expression was measured in FRT, FRT-ANO1, and PC-3 cells. (**C**) Effect of Ani-D1 and Ani-D2 on the protein expression levels of ANO1 in PC-3 cells. PC-3 cells were treated with 10 μM of Ani-D1 and Ani-D2 for 24 h. (Right) Summary of band intensity. The ANO1 band intensity was normalized to β-actin (mean ± S.E., *n* = 3). (**D**) ANO1 mRNA expression levels were determined by real-time PCR in PC-3 cells treated with Ani-D2 for 24 h (mean ± S.E., *n* = 3). ** *p* < 0.01.

**Figure 4 ijms-21-06470-f004:**
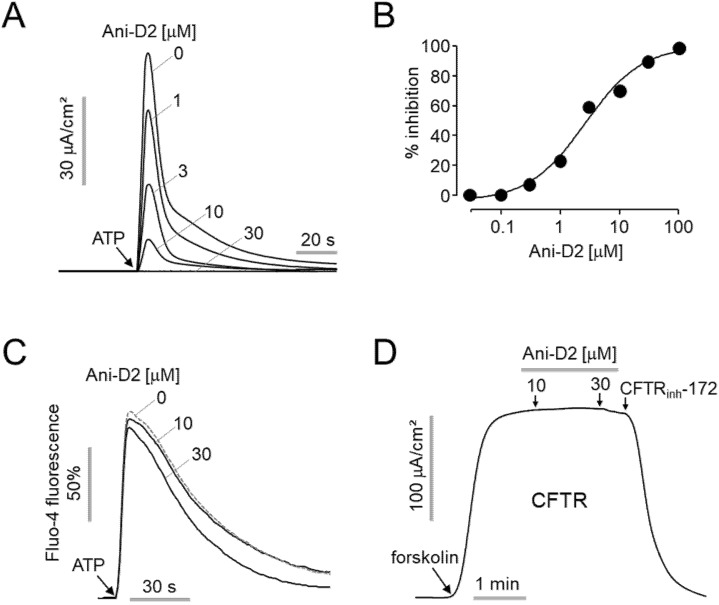
Characterization of Ani-D2. (**A**) Effect of Ani-D2 on apical membrane current was observed in FRT cells expressing ANO1. Ani-D2 was applied at the indicated concentrations 20 min prior to ANO1 activation by 100 µM ATP. (**B**) Summary of dose–responses (mean ± S.E., *n* = 3–4). (**C**) Effect of Ani-D2 on intracellular calcium concentration was measured by Fluo-4/NW in FRT cells. The cells were pretreated with 0, 10, 30 μM of Ani-D2 for 20 min prior to the treatment of 100 µM ATP. (**D**) Effect of Ani-D2 on CFTR chloride channel activity was observed in FRT cells expressing human CFTR. CFTR chloride currents were activated by 10 μM forskolin and inhibited by 10 μM CFTR_inh_-172.

**Figure 5 ijms-21-06470-f005:**
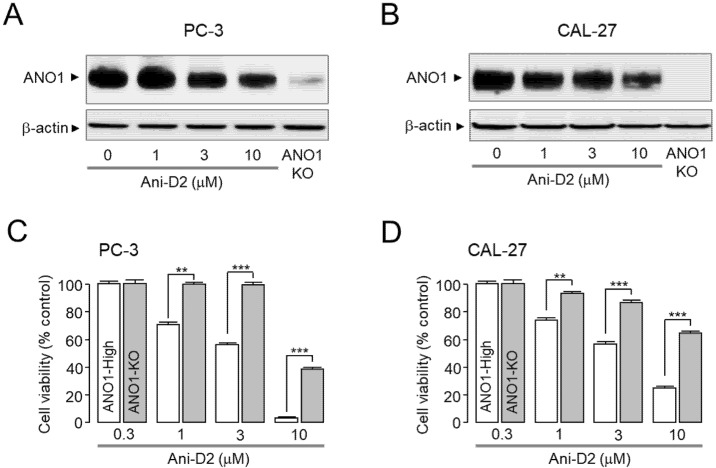
Effect of Ani-D2 on protein expression levels of ANO1 and cell viability in PC-3 and CAL-27 cells. (**A**,**B**) Representative Western blot analysis of ANO1 in Ani-D2 treated PC-3 and CAL-27 cells expressing ANO1 (three independent experiments performed). Cells were cultured with the indicated concentrations of Ani-D2 for 24 h. ANO1 knockout (KO) cells were established using CRISPR/Cas9 technique. (**C**,**D**) Effect of Ani-D2 on cell viability in PC-3, ANO1 KO PC-3, CAL-27, and ANO1 KO CAL-27 cells. Ani-D2 was treated at the indicated concentrations for 72 h, and cell viability was determined using MTS assay kit (mean ± S.E., *n* = 5). ** *p* < 0.01, *** *p* < 0.001.

**Figure 6 ijms-21-06470-f006:**
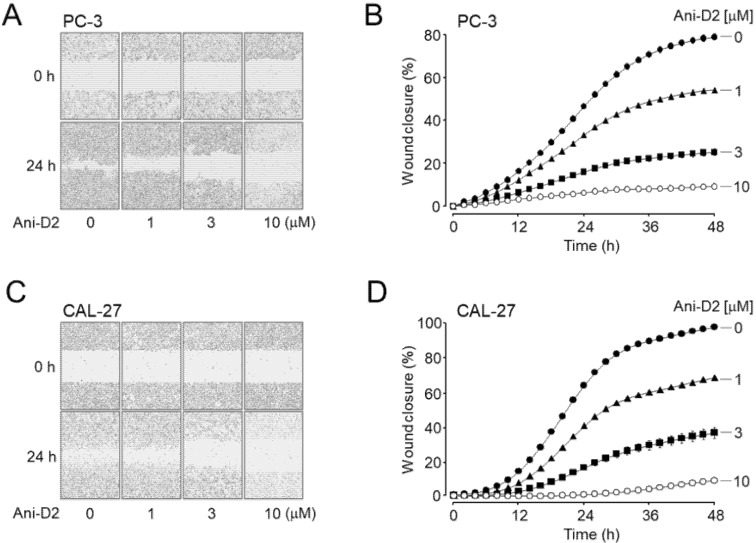
Effect of Ani-D2 on cell migration in PC-3 and CAL-27 cells. (**A**,**B**) Wound healing assay was performed in PC-3 cells. Cells were treated with Ani-D2 and representative images were taken at 0 and 24 h post wounding (×10). The wound closure was measured for 48 h (mean ± S.E., *n* = 3–4). (**C**,**D**) Wound healing assay was performed in CAL-27 cells. Cells were treated with Ani-D2 at the indicated concentration and representative images were taken at 0 and 24 h post wounding (×10). The wound closure was measured for 48 h (mean ± S.E., *n* = 3–4).

**Figure 7 ijms-21-06470-f007:**
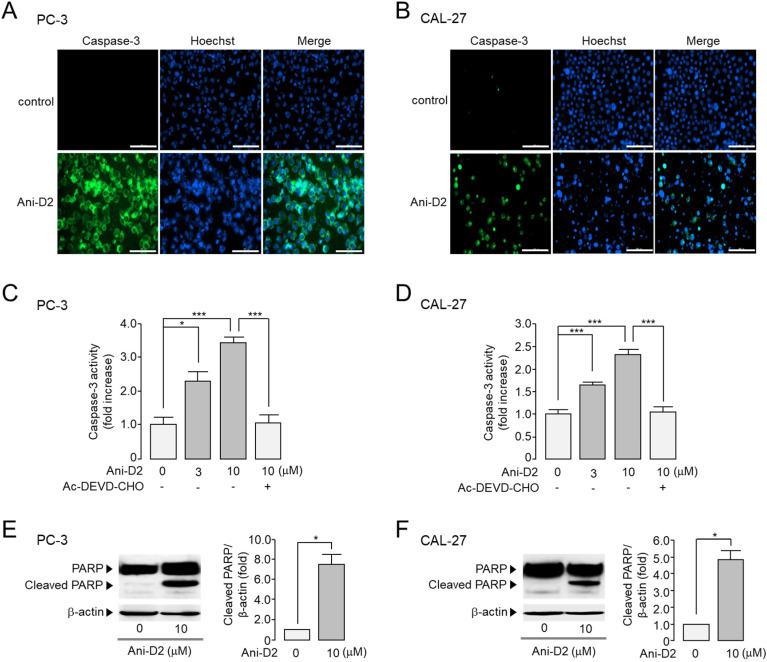
Effect of Ani-D2 on caspase-3 activity and cleavage of PARP in PC-3 and CAL-27 cells. (**A**,**B**) Images were taken after 24 h incubation with Ani-D2. Caspase-3 substrate (green, 2.0 μM) and Hoechst 33342 (blue, 1 μM) were treated for 20 min prior to image acquisition. The white bar represents 200 μm. (**C**,**D**) Cells were cultured with Ani-D2 at the indicated concentrations for 24 h, then 2 μM of caspase-3 substrate was treated for 20 min. Caspase-3 activity was inhibited by 10 μM of Ac-DEVD-CHO (mean ± S.E., *n* = 3–4). (**E**,**F**) Cells were cultured with 10 μM of Ani-D2 for 24 h, and expression level of PARP, cleaved PARP, and β-actin were measured by immunoblot analysis (mean ± S.E., *n* = 3). * *p* < 0.05, *** *p* < 0.001.

**Table 1 ijms-21-06470-t001:** ^1^H and ^13^C NMR spectroscopic data for compounds **1** and **2**.

C		1		2
	δ_C_ ^(a)^	δ_H_ ^(a)^ (mult., *J =* Hz)	δ_C_ ^(b)^	δ_H_ ^(b)^ (mult., *J =* Hz)
2	79.1	-	78.4	-
3	125.3	5.52 (d, 10.0)	126.3	5.50 (d, 10.5)
4	118.0	6.61 (d, 10.0)	115.3	6.62 (d, 10.5)
5	157.7	-	162.3	-
6	106.7	-	102.8	-
7	167.5	-	162.5	-
8	96.7	5.93 (s)	96.3	5.95 (s)
9	161.6	-	158.4	-
10	104.0	-	103.1	-
11	28.2	1.56 (s)	28.5	1.44 (s)
12	28.2	1.56 (s)	28.4	1.43 (s)
1′	194.1		196.1	
2′	125.5	8.00 (d, 15.5)	43.1	2.78 (dd, 3.0, 17.0) 3.07 (dd, 13.0, 17.0)
3′	143.7	7.70 (d, 15.5)	78.9	5.34 (dd, 3.0, 13.0)
4′	128.4	-	130.5	-
5′, 9′	131.2	7.52 (d, 8.5)	128.0	7.31 (d, 8.5)
6′, 8′	117.0	6.86 (d, 8.5)	115.7	6.87 (d, 8.5)
7′	161.2	-	161.2	-

^(a)^ recorded in CD_3_OD, ^(b)^ recorded in CDCl_3_; assignments were done by HSQC, HMBC, and COSY experiments (Appendix A).

**Table 2 ijms-21-06470-t002:** Cytotoxic effects of compounds **1**–**6** on human cancer cell line, PC-3.

Compounds	Cell Viability	IC_50_ (μM)
**1**	30.5 ± 0.28	8.89 ± 0.17
**2**	12.0 ± 0.11	7.29 ± 0.20
**3**	1.9 ± 0.02	1.60 ± 0.05
**4**	78.3 ± 0.72	>30
**5**	97.4 ± 0.90	>30
**6**	102.5 ± 0.94	>30
Capecitabine	27.1 ± 0.20	11.2 ± 1.44

PC-3 cells were treated with compounds at concentration of 30 μM for 48 h, and the cell viability was determined using MTS assay kit (mean ± S.E., *n* = 3).

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
