# Peer review of "Novel ANO1 Inhibitor from Mallotus apelta Extract Exerts Anticancer Activity through Downregulation of ANO1"

_ijms, 2020, doi:10.3390/ijms21186470_

Round 1
Reviewer 1 Report
The manuscript describes potentially important findings on anti-tumor effects of novel phytochemicals. Some essential controls are missing. The results would be worth publishing after including the experiments listed below.
Most importantly; What are the effects of the compounds in non-cancerous cells? Is the cytotoxicity at the same level?
Minor issues:
- The Introduction should describe the tissues where the ANO1 is normally expressed.
- Table 2 shows IC50 values. The inhibition curves where the IC50 values were calculated from should be shown (maybe in Supplementary Information).
- The discussion should mention that the cytotoxic effects of these novel compounds and other compounds tested earlier can also be explained by effects on some other cellular targets, not only on ANO1.
- The numbers of replicates in each experiment should be explained in Method section. In some results, the number of experiments has not been told.
- It is not said how many times the western blots were repeated.
Author Response
Reviewer #1:
Comments to the Author
The manuscript describes potentially important findings on anti-tumor effects of novel phytochemicals. Some essential controls are missing. The results would be worth publishing after including the experiments listed below.
- Most importantly; What are the effects of the compounds in non-cancerous cells? Is the cytotoxicity at the same level?
Response: Thank you for the valuable comments. To investigate the cytotoxic effect of compound 2 (Ani-D2) on non-cancerous cells, we observed the cytotoxicity of Ani-D2 in HaCaT, an immortal keratinocyte cell line spontaneously transformed from adult skin. Ani-D2 showed low toxicity in HaCaT cells compared with PC-3 and CAL-27 cells up to 3 mM, but high concentration of Ani-D2 (10 mM) showed strong reduction of cell viability in HaCaT cells (Figure S13). Previous study showed that ANO1 knockdown significantly reduced cell proliferation in both tumor cell lines and HaCaT cells. Notably, PC-3 and HaCaT cells expressed similar levels of ANO1 protein, but ANO1 knockdown more strongly reduced cell proliferation of PC-3 cells compared to HaCaT cells (Oncotarget 2016, 7(48):78619-78630). Our results are consistent with the previous report. Although further research is needed in the future, these results suggest that downregulation of ANO1 by small-molecule inhibitors may be more harmful to tumor cells than normal cells. This information has been described in the revised text.
- The Introduction should describe the tissues where the ANO1 is normally expressed.
Response: ANO1 is expressed in various tissues such as trachea, salivary gland, prostate, pancreas and colon. This information has been described in the revised text.
- Table 2 shows IC50 values. The inhibition curves where the IC50 values were calculated from should be shown (maybe in Supplementary Information).
Response: Thank you. The dose response curves for compounds 1, 2, 3 and capecitabine are shown in Figure S12.
- The discussion should mention that the cytotoxic effects of these novel compounds and other compounds tested earlier can also be explained by effects on some other cellular targets, not only on ANO1.
Response: Compound 1 and 2 are novel compounds and there are only few references for biological effects of benzopyran derivatives from M. apelta. As described in discussion, flavonoids extracted from M. apelta leaf have antioxidant activity and inhibit liver fibrosis in a carbon tetrachloride (CCl4)-induced fibrosis in rat via modulating TGF-beta/Smad and NF-κB signaling [32]. Some benzopyran derivatives from M. apelta show cytotoxic effects on human hepatocellular carcinoma (Hep-2) and rhabdosarcoma (RD) cells, but the underlying mechanism is not clear [33, 34].
- The numbers of replicates in each experiment should be explained in Method section. In some results, the number of experiments has not been told.
Response: Thank you. Corrected.
- It is not said how many times the western blots were repeated.
Response: Thank you. Corrected.
Reviewer 2 Report
In this paper, Seo et al. showed that a new compound extracted from a plant has activity to decrease steady state expression level of ANO1 and to induce apoptosis in cancer cells. The phenomenon to effect on ANO1 expression is intriguing, but its mechanistic study has not been performed. Furthermore, a relationship between downregulation of ANO1 and anti-cancer activity is not solid in this study. Thus this study is not suitable to International Journal of Molecular Sciences, and this would match to journals on medicinal chemistry or natural chemistry.
Comments:
- Ani-D2 does not become a keyword. Keywords should be general.
- As spectroscopic data of 1 and 2, optical rotation value is needed, since these have chirality.
- The section “2.2. Structural elucidations of compounds” is considered not to be required. These days, spectroscopic analysis is usual, and just putting the data in experimental section is enough.
- There is a data of anti-cell migration activity of compound (Figure 6). But in this concentration, cells die (Figure 5). Thus, it would not be anti-cell migration activity.
- Line 247: NF-kB should be NF-kappa(symbol)B.
Author Response
Reviewer #2:
Comments to the Author
In this paper, Seo et al. showed that a new compound extracted from a plant has activity to decrease steady state expression level of ANO1 and to induce apoptosis in cancer cells. The phenomenon to effect on ANO1 expression is intriguing, but its mechanistic study has not been performed. Furthermore, a relationship between downregulation of ANO1 and anti-cancer activity is not solid in this study. Thus this study is not suitable to International Journal of Molecular Sciences, and this would match to journals on medicinal chemistry or natural chemistry.
- Ani-D2 does not become a keyword. Keywords should be general.
Response: Thank you for your comment. The Ani-D2 has been replaced with ‘inhibitor’ in Keywords.
- As spectroscopic data of 1 and 2, optical rotation value is needed, since these have chirality.
Response: Compound 1 does not contain any stereo carbon. So its optical rotation cannot be measured. As mentioned in the main text, the optical rotation of 2 was approximate zero implying that it was isolated as a racemic mixture. This deduction was in line with no Cotton effect observed in the ECD spectrum of 2. The structure of 2 was established based on those data and named malloapelta II. Although we tried to separate the enantiomers of compound 2 using several chiral columns, it could not be accomplished due to the limited amount of compound 2.
- The section “2.2. Structural elucidations of compounds” is considered not to be required. These days, spectroscopic analysis is usual, and just putting the data in experimental section is enough.
Response: The authors understand the reviewer’s opinion. In the case of known compounds, we just added and cited it in the manuscript. To ensure the structure is correct, we want to keep the elucidation description of the two new compounds in Section 2.2.
- There is a data of anti-cell migration activity of compound (Figure 6). But in this concentration, cells die (Figure 5). Thus, it would not be anti-cell migration activity.
Response: The authors agree with the reviewer’s comment. It is well known that downregulation of ANO1 significantly reduces cell migration activity in several cancer cells including prostate cancer and head and neck squamous cell carcinoma (HNSCC). However, it is hard to remove the cytotoxic effect of ANO1 inhibitors when we observed the effect of ANO1 inhibitor on cell migration because ANO1 inhibition affects both cell proliferation and migration. In the scratch wound healing assay, the cells were wounded after they reached 100% confluence, and the cells were maintained with 2% FBS containing medium to minimize the cell proliferation effect. In both PC-3 and CAL-27 cells, there is small difference in the inhibition rate of cell viability when cells were treated with 1 and 3 µM Ani-D2 (Figure 5), however the difference in the inhibition rate of wound healing activity between 1 and 3 µM Ani-D2 was significantly increased (Figure 6). We believe that these results showing anti-migration effect of Ani-D2.
- Line 247: NF-kB should be NF-kappa(symbol)B.
Response: Thank you. Corrected.
Reviewer 3 Report
The manuscript entitled “Novel ANO1 Inhibitor from Mallotus apelta Extract Exerts Anticancer Activity through Downregulation 3 of ANO1” describes the identification of new compounds capable to inhibit ANO1 and to downregulate its expression. In a whole, the article is clear and the text well written. However, the authors need to check the comments listed below:
Comment 1:
There is no rationale explaining why the authors have first evaluated the cytotoxic activity of the purified compounds while they are looking for new ANO-1 inhibitors (see the introductive section and the discussion lines 233-234). The link between sections 2.3 and 2.4 is clearly not obvious. Why did the authors think that the cytotoxic activity of 1 and 2 might be due to their ability of inhibiting ANO1? Cytotoxicity can originate from many different sources. So, if the authors are looking for new ANO1 inhibitors, why don’t they use first their FRT cells-based assay (Figure 3A)? To me, sections 2.3 and 2.4 should be reorganized to make the story clear.
Comment 2:
The justification why the authors stop studying compound 1 is not satisfying. First, this compound would be a good control to determine whether or not the loss of ANO1 expression is an important determinant of the anticancer activity of 2. Second, 1 shows a cytotoxic activity and inhibits ANO-1 in vitro. So, it would be very interesting to determine whether this compound might exert its anticancer activity solely by inhibiting ANO-1 channel activity. This would bring novel insights in comparison with other compounds (e.g. kaempferol and Ani9) that inhibit ANO1 channel activity without affecting cancer cell viability. Additional experiments (ANO1 KO cells, migration) should be performed with 1. This will help clarifying the link between ANO1 inhibition, expression and anticancer activity.
Comment 3:
The authors state that “…blockade of ANO1 inhibits cell proliferation of metastatic prostate cancer cells…”. This suggests that ANO1 might be considered as a driver component of oncogenesis. However, when the authors genetically delete ANO1 in either PC-3 or CAL-27 cells, it seems that the complete loss of ANO1 expression does not interfere with the viability of the two cell lines (see Figures 5C and 5D). I do perfectly agree that it is interesting to see that 2 has no effect on ANO1 KO cells. However, I would have expected that untreated ANO1 KO cells would have been less fit than their ANO1 expressing counterpart. This is not what is shown here. So, how do the authors explain this discrepancy?
Comment 4:
If the loss of ANO1 expression is an important feature of compound 2 activity, it would be interesting to determine at which level (transcriptional, translational, degradation) it acts. The authors should at least determine whether this loss is due to a decreased expression of the coding gene. If this is not the case, post-transcriptional events are likely to be involved.
Minor comment 1:
The Glucose molecule should be deleted from Figure 1.
Minor comment 2:
The complete loss of ANO1 expression in ANO1 KO cells should be confirmed by Western blot.
Minor comment 3:
The last paragraph of the discussion section (lines 269-278) should be deleted since it only repeats what is indicated above.
Minor comment 4:
Line 166: the sentence “Pretreatment Ani-D2 significantly inhibited ATP-activated ANO1 chloride…” should be read “Pretreatment with Ani-D2 significantly inhibited ATP-activated ANO1 chloride…”.
Minor comment 5:
Line 187: “interestingly” should be removed.
Minor comment 6:
Reference 34 should be modified accordingly to MDPI recommendations.
Author Response
Reviewer #3:
Comments to the Author
The manuscript entitled “Novel ANO1 Inhibitor from Mallotus apelta Extract Exerts Anticancer Activity through Downregulation 3 of ANO1” describes the identification of new compounds capable to inhibit ANO1 and to downregulate its expression. In a whole, the article is clear and the text well written. However, the authors need to check the comments listed below:
- There is no rationale explaining why the authors have first evaluated the cytotoxic activity of the purified compounds while they are looking for new ANO-1 inhibitors (see the introductive section and the discussion lines 233-234). The link between sections 2.3 and 2.4 is clearly not obvious. Why did the authors think that the cytotoxic activity of 1 and 2 might be due to their ability of inhibiting ANO1? Cytotoxicity can originate from many different sources. So, if the authors are looking for new ANO1 inhibitors, why don’t they use first their FRT cells-based assay (Figure 3A)? To me, sections 2.3 and 2.4 should be reorganized to make the story clear.
Response: Sorry for the confusion. When we started this project, we just wanted to investigate the cytotoxic effects of the compounds on cancer cells, and we found out novel compounds (1 and 2) have cytotoxic effect. Next, we observed the effects of the compounds on ANO1 activity because ANO1 assay was available at the time, and ANO1 is a promising anticancer target protein. Fortunately, we could found out the novel compounds (1 and 2) are bona-fide ANO1 inhibitors. In conclusion, all authors agree with the reviewer’s opinion. Reorganization of sections 2.3 and 2.4 will make the story clear. The section 2.3 and 2.4 were reorganized in the revised manuscript.
- The justification why the authors stop studying compound 1 is not satisfying. First, this compound would be a good control to determine whether or not the loss of ANO1 expression is an important determinant of the anticancer activity of 2. Second, 1 shows a cytotoxic activity and inhibits ANO-1 in vitro. So, it would be very interesting to determine whether this compound might exert its anticancer activity solely by inhibiting ANO-1 channel activity. This would bring novel insights in comparison with other compounds (e.g. kaempferol and Ani9) that inhibit ANO1 channel activity without affecting cancer cell viability. Additional experiments (ANO1 KO cells, migration) should be performed with 1. This will help clarifying the link between ANO1 inhibition, expression and anticancer activity.
Response: Thank you for the comments. We were not interested in compound 1 because compound 1 is weakly decreased ANO1 protein levels (compound 1 decreased ANO1 more strongly at high concentration, data not shown) compared to compound 2. In previous studies [13, 35], we had shown that ANO1 inhibitors affect both channel activity and protein expression levels of ANO1 have more potent anticancer activities compared to the ANO1 inhibitors that only inhibits ANO1 channel activity. Thus, we focused on the anticancer effects of compound 2 because it inhibited ANO1 channel activity and reduced ANO1 protein expression levels. As shown in Figure 5, at low concentrations of Ani-D2 (1 and 3 mM) did not strongly affect cell viability of ANO1-KO cells, however Ani-D2 strongly decreased cell viability at 10 mM in ANO1-KO PC-3 and CAL-27 cells. These results suggest that there is a mechanism of action other than ANO1 downregulation. This information is described in the discussion section.
- The authors state that “…blockade of ANO1 inhibits cell proliferation of metastatic prostate cancer cells…”. This suggests that ANO1 might be considered as a driver component of oncogenesis. However, when the authors genetically delete ANO1 in either PC-3 or CAL-27 cells, it seems that the complete loss of ANO1 expression does not interfere with the viability of the two cell lines (see Figures 5C and 5D). I do perfectly agree that it is interesting to see that 2 has no effect on ANO1 KO cells. However, I would have expected that untreated ANO1 KO cells would have been less fit than their ANO1 expressing counterpart. This is not what is shown here. So, how do the authors explain this discrepancy?
Response: As reviewer’s expectation, it is well known that transient ANO1 knockdown with siRNA significantly decrease cell proliferation and migration in several cancer cells expressing high levels of ANO1. However, in this study, we used stable ANO1 knockout cell lines. We established ANO1 knockout cell line using ANO1 CRISPR/Cas9, and cells that grow well without ANO1 were selected during the selection period. Thus, ANO1-KO PC-3 and CAL-27 cells may be different from the original PC-3 and CAL-27 cells expressing high levels of ANO1. There is no significant difference in doubling time between original and KO cell line. The ANO1 KO cell line is not a perfect model but good for use to investigate whether a compound effect on cell viability or migration through ANO1-dependent manner.
- If the loss of ANO1 expression is an important feature of compound 2 activity, it would be interesting to determine at which level (transcriptional, translational, degradation) it acts. The authors should at least determine whether this loss is due to a decreased expression of the coding gene. If this is not the case, post-transcriptional events are likely to be involved.
Response: Thank you for the helpful comments. We performed quantitative real-time PCRs to investigate whether Ani-D2 affects the mRNA expression of ANO1 in PC-3 cells. As shown in Figure 3D, Ani-D2 did not affect the ANO1 mRNA expression up to 10 µM. In addition, Ani-2D directly blocks ANO1 channel activity without affecting intracellular calcium signaling (Figure 4). These results suggest that Ani-D2 reduces the protein stability of ANO1.
- The Glucose molecule should be deleted from Figure 1.
Response: Thank you. Corrected.
- The complete loss of ANO1 expression in ANO1 KO cells should be confirmed by Western blot.
Response: Sorry for the confusion. ANO1 expressions in ANO1 KO cells were confirmed by Western blot in Figure 5A and 5B. Please find the rightmost well of each blot.
- The last paragraph of the discussion section (lines 269-278) should be deleted since it only repeats what is indicated above.
Response: Thank you. Corrected.
- Line 166: the sentence “Pretreatment Ani-D2 significantly inhibited ATP-activated ANO1 chloride…” should be read “Pretreatment with Ani-D2 significantly inhibited ATP-activated ANO1 chloride…”.
Response: Thank you. Corrected.
- Line 187: “interestingly” should be removed.
Response: Thank you. Corrected.
- Reference 34 should be modified accordingly to MDPI recommendations.
Response: Thank you. Corrected.
Round 2
Reviewer 1 Report
Thank you for responding to the comments. Please check the English language once more as there were still some errors.
Author Response
We greatly appreciate the editor’s and reviewers’ efforts to carefully review our manuscript and the valuable comments and suggestions offered for the improvement of the manuscript (ijms-888983). We have made each of the suggested revisions and the points of criticism raised by the reviewers were addressed by a point-by-point response. Changes in the manuscript text are highlighted in red color font.
Reviewer #1:
Comments to the Author
Thank you for responding to the comments. Please check the English language once more as there were still some errors.
Response: Thank you. We have carefully revised the manuscript.
Reviewer 3 Report
The authors have responded to some of my comments but still some modifications and clarifications are required. See below:
Initial comment 1:
There is no rationale explaining why the authors have first evaluated the cytotoxic activity of the purified compounds while they are looking for new ANO-1 inhibitors (see the introductive section and the discussion lines 233-234). The link between sections 2.3 and 2.4 is clearly not obvious. Why did the authors think that the cytotoxic activity of 1 and 2 might be due to their ability of inhibiting ANO1? Cytotoxicity can originate from many different sources. So, if the authors are looking for new ANO1 inhibitors, why don’t they use first their FRT cells-based assay (Figure 3A)? To me, sections 2.3 and 2.4 should be reorganized to make the story clear.
Response: Sorry for the confusion. When we started this project, we just wanted to investigate the cytotoxic effects of the compounds on cancer cells, and we found out novel compounds (1 and 2) have cytotoxic effect. Next, we observed the effects of the compounds on ANO1 activity because ANO1 assay was available at the time, and ANO1 is a promising anticancer target protein. Fortunately, we could found out the novel compounds (1 and 2) are bona-fide ANO1 inhibitors. In conclusion, all authors agree with the reviewer’s opinion. Reorganization of sections 2.3 and 2.4 will make the story clear. The section 2.3 and 2.4 were reorganized in the revised manuscript.
New comment 1: Reorganization of sections 2.3 and 2.4 has been done. However, the abstract should be changed accordingly. So, the sentences “These compounds were evaluated for their cytotoxic effects on PC-3 prostate cancer cells and their inhibitory effects on ANO1 channel activity. Interestingly, compounds 1 and 2 significantly reduced both cell viability of PC-3 cells and ANO1 channel activity.” (lines 35-37) should be changed to “These compounds were evaluated for their inhibitory effects on ANO1 channel activity and their cytotoxic effects on PC-3 prostate cancer cells. Interestingly, compounds 1 and 2 significantly reduced both ANO1 channel activity and cell viability.” Moreover, the entire section 2.4 (lines 180-186) should be placed before the paragraph going from lines 160 to 169. This is indeed after the cell viability assay, the authors decided to only study compound 2.This paragraph (lines 160-169) might then become “2.5 Selective activity of Ani-D2 on ANO1”.
Initial comment 2:
The justification why the authors stop studying compound 1 is not satisfying. First, this compound would be a good control to determine whether or not the loss of ANO1 expression is an important determinant of the anticancer activity of 2. Second, 1 shows a cytotoxic activity and inhibits ANO-1 in vitro. So, it would be very interesting to determine whether this compound might exert its anticancer activity solely by inhibiting ANO-1 channel activity. This would bring novel insights in comparison with other compounds (e.g. kaempferol and Ani9) that inhibit ANO1 channel activity without affecting cancer cell viability. Additional experiments (ANO1 KO cells, migration) should be performed with 1. This will help clarifying the link between ANO1 inhibition, expression and anticancer activity.
Response: Thank you for the comments. We were not interested in compound 1 because compound 1 is weakly decreased ANO1 protein levels (compound 1 decreased ANO1 more strongly at high concentration, data not shown) compared to compound 2. In previous studies [13, 35], we had shown that ANO1 inhibitors affect both channel activity and protein expression levels of ANO1 have more potent anticancer activities compared to the ANO1 inhibitors that only inhibits ANO1 channel activity. Thus, we focused on the anticancer effects of compound 2 because it inhibited ANO1 channel activity and reduced ANO1 protein expression levels. As shown in Figure 5, at low concentrations of Ani-D2 (1 and 3 mM) did not strongly affect cell viability of ANO1-KO cells, however Ani-D2 strongly decreased cell viability at 10 mM in ANO1-KO PC-3 and CAL-27 cells. These results suggest that there is a mechanism of action other than ANO1 downregulation. This information is described in the discussion section.
New comment 2: the response of the authors is still not satisfying to me. The authors assume that compound 1 is not interesting because it does not act on ANO1 expression levels. To make sure compound 1 exert its cytotoxic activity independently of its ability to inhibit ANO1 activity, the authors must assess its effect on ANO1 KO cells. If compound 1 still inhibits cell viability in absence of ANO1 expression, then, the authors can justify why they stop studying it further. Otherwise, the justification is only based on an assumption made by the authors on what was shown for kaempferol or Ani9. However, in contrast to compound 1, kaempferol or Ani9 had a weak effect on cell viability.
Initial comment 3:
The authors state that “…blockade of ANO1 inhibits cell proliferation of metastatic prostate cancer cells…”. This suggests that ANO1 might be considered as a driver component of oncogenesis. However, when the authors genetically delete ANO1 in either PC-3 or CAL-27 cells, it seems that the complete loss of ANO1 expression does not interfere with the viability of the two cell lines (see Figures 5C and 5D). I do perfectly agree that it is interesting to see that 2 has no effect on ANO1 KO cells. However, I would have expected that untreated ANO1 KO cells would have been less fit than their ANO1 expressing counterpart. This is not what is shown here. So, how do the authors explain this discrepancy?
Response: As reviewer’s expectation, it is well known that transient ANO1 knockdown with siRNA significantly decrease cell proliferation and migration in several cancer cells expressing high levels of ANO1. However, in this study, we used stable ANO1 knockout cell lines. We established ANO1 knockout cell line using ANO1 CRISPR/Cas9, and cells that grow well without ANO1 were selected during the selection period. Thus, ANO1-KO PC-3 and CAL-27 cells may be different from the original PC-3 and CAL-27 cells expressing high levels of ANO1. There is no significant difference in doubling time between original and KO cell line. The ANO1 KO cell line is not a perfect model but good for use to investigate whether a compound effect on cell viability or migration through ANO1-dependent manner.
New comment 3: This issue should be mentioned and discussed in the discussion section.
Additional comment 1:
The contributions of the new listed author Dongkyu Jeon are not indicated in the dedicated section.
Additional comment 2:
The meaning of FRT (Fisher Rat Thyroid) should be indicated the first time it appears in the text (line 143, not 161).
Additional comment 3:
Line 200: the fact that the HaCat cells express ANO1 should be indicated here, not only in the discussion section.
Additional comment 4:
The last paragraph of the discussion section (lines 279-291) should be placed before the preceding one (lines 271-278).
Author Response
We greatly appreciate the editor’s and reviewers’ efforts to carefully review our manuscript and the valuable comments and suggestions offered for the improvement of the manuscript (ijms-888983). We have made each of the suggested revisions and the points of criticism raised by the reviewers were addressed by a point-by-point response. Changes in the manuscript text are highlighted in red color font.
Reviewer #3:
Comments to the Author
New comment 1. Reorganization of sections 2.3 and 2.4 has been done. However, the abstract should be changed accordingly. So, the sentences “These compounds were evaluated for their cytotoxic effects on PC-3 prostate cancer cells and their inhibitory effects on ANO1 channel activity. Interestingly, compounds 1 and 2 significantly reduced both cell viability of PC-3 cells and ANO1 channel activity.” (lines 35-37) should be changed to “These compounds were evaluated for their inhibitory effects on ANO1 channel activity and their cytotoxic effects on PC-3 prostate cancer cells. Interestingly, compounds 1 and 2 significantly reduced both ANO1 channel activity and cell viability.” Moreover, the entire section 2.4 (lines 180-186) should be placed before the paragraph going from lines 160 to 169. This is indeed after the cell viability assay, the authors decided to only study compound 2. This paragraph (lines 160-169) might then become “2.5 Selective activity of Ani-D2 on ANO1”.
Response: We sincerely appreciate reviewer’s thoughtful comments and suggestions. We have revised the manuscript as reviewer’s suggestions.
New comment 2. The response of the authors is still not satisfying to me. The authors assume that compound 1 is not interesting because it does not act on ANO1 expression levels. To make sure compound 1 exert its cytotoxic activity independently of its ability to inhibit ANO1 activity, the authors must assess its effect on ANO1 KO cells. If compound 1 still inhibits cell viability in absence of ANO1 expression, then, the authors can justify why they stop studying it further. Otherwise, the justification is only based on an assumption made by the authors on what was shown for kaempferol or Ani9. However, in contrast to compound 1, kaempferol or Ani9 had a weak effect on cell viability.
Response: Thank you for the reviewer’s thoughtful comments. As reviewer’s opinion, there is a possibility that compound 1 may be different from kaempferol or Ani9, but unfortunately, the experiment cannot be performed with compound 1 because the amount of compound 1 is not sufficient and it takes too much time to purify compound 1. Compounds 1 and 2 are very similar in structure, and compound 2 shows non-specific cytotoxicity at high concentration (10 mM) in both ANO1-KO PC-2 and CAL-27 cells, so compound 1 is also expected to exhibit non-specific cytotoxicity at high concentration. In general, it is hard to separate the inhibitory effect of a compound on ANO1 from its non-specific effects. Compound 2 showed good properties for specific regulation of ANO1 channel activity and protein levels at low concentrations, so we focused on compound 2 in this study. We are sorry for not being able to perform the suggested experiments, but our experiments with compound 2 could provide useful information for studying ANO1.
New comment 3: This issue should be mentioned and discussed in the discussion section.
Response: Thank you. The issue has been described in the discussion section.
Additional comment 1: The contributions of the new listed author Dongkyu Jeon are not indicated in the dedicated section.
Response: Thank you. Corrected.
Additional comment 2: The meaning of FRT (Fisher Rat Thyroid) should be indicated the first time it appears in the text (line 143, not 161).
Response: Thank you. Corrected.
Additional comment 3: Line 200: the fact that the HaCat cells express ANO1 should be indicated here, not only in the discussion section.
Response: Thank you. Corrected.
Additional comment 4: The last paragraph of the discussion section (lines 279-291) should be placed before the preceding one (lines 271-278).
Response: Thank you. Corrected.